# Corin: A Key Mediator in Sodium Homeostasis, Vascular Remodeling, and Heart Failure

**DOI:** 10.3390/biology11050717

**Published:** 2022-05-07

**Authors:** Xianrui Zhang, Xiabing Gu, Yikai Zhang, Ningzheng Dong, Qingyu Wu

**Affiliations:** 1Cyrus Tang Hematology Center, Collaborative Innovation Center of Hematology, State Key Laboratory of Radiation Medicine and Prevention, Soochow University, Suzhou 215123, China; zhangxianrui94@126.com (X.Z.); xiabinggoouu@163.com (X.G.); 20194253022@stu.suda.edu.cn (Y.Z.); ningzhengdong@suda.edu.cn (N.D.); 2MOH Key Laboratory of Thrombosis and Hemostasis, Jiangsu Institute of Hematology, The First Affiliated Hospital of Soochow University, Suzhou 215006, China

**Keywords:** apical membrane trafficking, atrial natriuretic peptide, corin, eccrine sweat glands, heart failure, protease, renal epithelial cells, sodium homeostasis, spiral artery remodeling

## Abstract

**Simple Summary:**

Atrial natriuretic peptide (ANP) is an important hormone that regulates many physiological and pathological processes, including electrolyte and body fluid balance, blood volume and pressure, cardiac channel activity and function, inflammatory response, lipid metabolism, and vascular remodeling. Corin is a transmembrane serine protease that activates ANP. Variants in the *CORIN* gene are associated with cardiovascular disease, including hypertension, cardiac hypertrophy, atrial fibrillation, heart failure, and preeclampsia. The current data indicate a key role of corin-mediated ANP production and signaling in the maintenance of cardiovascular homeostasis. In this review, we discuss the latest findings regarding the molecular and cellular mechanisms underlying the role of corin in sodium homeostasis, uterine spiral artery remodeling, and heart failure.

**Abstract:**

Atrial natriuretic peptide (ANP) is a crucial element of the cardiac endocrine function that promotes natriuresis, diuresis, and vasodilation, thereby protecting normal blood pressure and cardiac function. Corin is a type II transmembrane serine protease that is highly expressed in the heart, where it converts the ANP precursor to mature ANP. Corin deficiency prevents ANP activation and causes hypertension and heart disease. In addition to the heart, corin is expressed in other tissues, including those of the kidney, skin, and uterus, where corin-mediated ANP production and signaling act locally to promote sodium excretion and vascular remodeling. These results indicate that corin and ANP function in many tissues via endocrine and autocrine mechanisms. In heart failure patients, impaired natriuretic peptide processing is a common pathological mechanism that contributes to sodium and body fluid retention. In this review, we discuss most recent findings regarding the role of corin in non-cardiac tissues, including the kidney and skin, in regulating sodium homeostasis and body fluid excretion. Moreover, we describe the molecular mechanisms underlying corin and ANP function in supporting orderly cellular events in uterine spiral artery remodeling. Finally, we assess the potential of corin-based approaches to enhance natriuretic peptide production and activity as a treatment of heart failure.

## 1. Introduction

The cardiac natriuretic peptides function as a hormonal mechanism to regulate body fluid and electrolyte balance, thereby maintaining normal blood volume and pressure [1,2,3,4]. Deficiency in atrial natriuretic peptide (ANP) causes salt-sensitive hypertension in mice [5]. Genetic variants in the human *NPPA* gene, encoding ANP, have been identified as key determinants in blood pressure levels in large populations [1,6]. More recent studies have implicated the natriuretic peptides in other physiological and pathological processes, such as cardiac potassium channel activity [7], vascular remodeling [8,9], inflammatory response [10,11,12,13], and lipid metabolism [14,15,16].

The natriuretic peptides are synthesized as pre-pro-peptides. Post-translational modifications, including proteolytic processing and glycosylation, are important in regulating natriuretic peptide activities [17,18,19,20,21]. Particularly, cleavage of the pro-fragment is essential for the activation of these peptides. Corin was cloned from the human heart as a novel serine protease that includes a transmembrane domain near the N-terminus and multiple modules in the extracellular region [22] (Figure 1). Such a protein modular arrangement resembles those in other type II transmembrane serine proteases, a group of trypsin-like enzymes involved in diverse biological processes [23,24]. Biochemical and genetic studies have shown that corin is the long-sought protease for ANP activation [20]. Corin also processes pro-brain or B-type natriuretic peptide (pro-BNP) in vitro [25,26]. Furin, however, likely plays a more important role in processing pro-BNP and pro-C-type natriuretic peptides (pro-CNP) [27,28].

To date, we have gained extensive knowledge regarding the biology of corin, including its genetics, biochemistry, enzymology, cell biology, and regulation, as described in previous reviews [20,29,30,31]. Variants in the *CORIN* gene that reduce corin expression and/or function have been described in hypertensive individuals [32,33,34,35,36,37]. In this review, we focus on recent findings of corin function in regulating sodium homeostasis and cardiovascular pathophysiology, particularly those published in the last three years. We will examine the function of corin in non-cardiac tissues, including those of the kidney and skin. Moreover, we will discuss newly uncovered molecular mechanisms underlying corin and ANP activity in promoting spiral artery remodeling in the pregnant uterus. Finally, we will evaluate the therapeutic potential of corin in heart failure.

## 2. Regulation of Sodium Homeostasis in Non-Cardiac Tissues

Sodium and body fluid homeostasis is crucial for normal blood pressure [38]. The heart is the major organ that produces ANP and BNP [2]. When blood volume and/or pressure increases, the heart releases the natriuretic peptides to enhance vasodilation in the peripheral tissue, and natriuresis and diuresis in the kidney. This cardiac endocrine function serves as a cardiorenal feedback mechanism to maintain normal blood volume and pressure [1,2,3,4].

Consistent with its role in processing cardiac natriuretic peptides, corin expression is most abundant in the heart [22]. The cardiac *CORIN* expression involves T-box transcription factor 5 (TBX5), GATA binding protein 4 (GATA4), and NK2 homeobox 5 (NKX2-5) transcription factors [39,40,41]. In single-cell RNA sequencing analyses, corin has been identified as an early surface marker that can be used to purify cardiomyocytes from human embryonic and induced pluripotent stem cells [41,42]. In corin knockout (KO) mice, ANP activation in the heart was undetectable [43], demonstrating the importance of corin in ANP generation.

In non-cardiac tissues, including the kidney and skin, corin expression has been detected [22,26,44,45]. Unlike secreted proteases, such as trypsin and prothrombin, corin is a transmembrane protein [22,46]. The single-span transmembrane domain at the N-terminus anchors corin onto the cell surface at the expression site. Based on this structural feature, corin is expected to function locally in the kidney and skin, as discussed below.

### 2.1. Renal Corin

#### 2.1.1. Expression in Renal Epithelial Cells

Kidneys in mice, rats, and humans have been shown to express corin mRNA and protein [22,26,47,48]. Similar renal expression has been reported for pro-ANP/ANP and the ANP receptor, natriuretic peptide receptor-A (NPR-A) [49]. Among renal segments, corin level is highest in the proximal tubules. The collecting ducts in the medulla express lower levels of corin, pro-ANP, and NPR-A. In contrast, little or no corin expression was detected in the glomerulus and the distal tubules [49].

The proximal tubules are important in renal reabsorption, a key physiological process in electrolyte, body fluid, and metabolic homeostasis [50]. The proximal tubule consists of a single layer of polarized epithelial cells connected by tight junctions. In experiments with confocal and electron microscopy, corin was found on the apical, but not basolateral, membrane in the polarized renal epithelial cells, which differs from the entire cell membrane distribution pattern in non-polarized cardiomyocytes [47,49,51]. The specific apical membrane expression probably indicates a function of corin on the lumen of the proximal tubules to inhibit sodium reabsorption, thereby promoting natriuresis. These findings also raise the question regarding the molecular mechanism underlying the apical corin expression in polarized epithelial cells.

Specific apical distribution is one of the distinct features of polarized epithelial cells. Impaired protein trafficking to the apical membrane in renal epithelial cells has been associated with kidney disease [52]. Protein structural elements, including the transmembrane domain, N- and O-glycans, glycosyl-phosphatidylinositol anchor, and amino acid motifs, are known sorting signals in apical trafficking [53]. A recent study identified a novel DSSDE motif in low-density lipoprotein receptor-like repeat 8 (LDLR8) of corin as an apical sorting signal in polarized renal epithelial cells [51]. Amino acid substitutions in this motif abolished the specific apical corin trafficking in polarized Madin–Darby canine kidney (MDCK) cells [51].

LDLR-like repeats are protein modules found in many secreted and cell surface proteins [54]. The DSSDE motif is also present in other LDLR-containing cell surface receptors on the apical membrane in renal proximal tubules. CD320, for example, is a transcobalamin receptor for vitamin B12 uptake [55,56]. The N-terminal extracellular region of CD320 contains two LDLR domains, the second of which has a DSSDE motif (Figure 2). In polarized renal and intestinal epithelial cells, the DSSDE motif is required for specific apical trafficking of CD320 [51,57]. These findings indicate that the DSSDE motif in LDLR repeats may have a general role in apical trafficking in polarized epithelial cells.

Additional studies have shown that the DSSDE motif-dependent apical trafficking of corin and CD320 is mediated by Rab11a [51,57], a member of the small GTPase superfamily, which plays a central role in apical trafficking in polarized epithelial cells [58] (Figure 2). Inhibition of Rab11a expression by a dominant negative Rab11a mutant or *RAB11A* gene knockdown abolished the specific apical targeting of corin and CD320 in MDCK and colon-derived Caco-2 cells [51,57]. Currently, it is unclear how Rab11a recognizes the DSSDE motif in corin and CD320 LDLR repeats. Further studies will be important to define the Rab11a-mediated mechanism, and to examine whether the DSSDE motif in other LDLR-containing proteins has a similar role in apical trafficking in polarized epithelial cells.

#### 2.1.2. Functional Significance of Renal Corin Expression

Electrolyte homeostasis is imperative for survival in all animals. In primitive vertebrates in salty water, natriuretic peptides act as a key mechanism to excrete excessive salt [59,60]. Corin is conserved in all vertebrate species, ranging from fish to mammals, indicating the importance of corin function in physiological homeostasis. In corin-null mice, urinary sodium excretion was reduced, especially on high-salt diets [61]. Previously, ANP was shown to function in the inner medullary collecting ducts to inhibit sodium absorption [62,63,64]. In the kidney, however, most solutes in the glomerular filtrate are absorbed in the proximal tubules. The findings of high levels of corin, ANP, and NPR-A expression in the proximal tubules suggest a corin and ANP-mediated autocrine mechanism in this key renal segment to inhibit salt and water reabsorption.

In agreement with a renal corin function in sodium homeostasis, increased renal corin levels have been observed in rats and humans on high-salt diets, possibly indicating a compensatory response to increase sodium excretion [65]. Another recent study in humans identified an association between *CORIN* variants and salt sensitivity, longitudinal blood pressure changes, and hypertension incidence [66,67]. In rat models of kidney injury and severe cardiorenal syndrome, low levels of renal corin were reported [47,68]. Similarly, low levels of renal corin were observed in patients with chronic kidney disease and sodium retention [48]. These findings indicate that impaired renal corin expression and/or function may be a pathological mechanism underlying sodium retention in kidney disease. Interestingly, cardiac and renal *Corin* gene expression responded differently in hypertensive rats treated with anti-hypertensive drugs [69]. Additional studies are needed to define the renal corin function, and to understand how cardiac and renal corin-mediated mechanisms are coordinated in regulating sodium and body fluid homeostasis. These studies may provide new insights into the pathogenesis of sodium and body fluid retention in patients with heart and kidney diseases.

### 2.2. Skin Corin

#### 2.2.1. Eccrine Sweat Glands

Overheating can be life-threatening. Sweating is a basic skin function to lower body temperature, which is mediated primarily by eccrine sweat glands. Among mammals, humans have the highest number of eccrine sweat glands on the skin surface [70]. This anatomical feature provides an evolutionary advantage for humans to survive in hot environments.

The original sweat produced in the eccrine glands is isotonic to plasma, with high levels of salt. Considerable amounts of salt are reabsorbed before sweat reaches the skin surface. This reabsorption process prevents salt loss and electrolyte imbalance, especially when large amounts of sweat is produced, for example, during hard labor or sports. To date, the molecular mechanisms controlling salt excretion and reabsorption in eccrine sweat glands are not completely understood [71,72].

Recently, corin, ANP, and NPR-A proteins were detected in the luminal epithelial cells in human and mouse eccrine sweat glands, indicating a potential function of corin and ANP in sweating [73]. Indeed, low levels of sweat and salt excretion were found in corin KO mice on normal- and high-salt diets, compared to those in wild-type (WT) mice [73]. When corin KO mice were treated with amiloride, an epithelial sodium channel (ENaC) inhibitor, sweat and salt excretion was normalized. Importantly, reduced sweat and salt excretion was not found in corin conditional KO mice, i.e., mice lacking only cardiac corin. These results indicate that corin-mediated ANP production and signaling in the skin promote sweat and salt excretion by inhibiting ENaC, which mediates sodium reabsorption in the eccrine sweat ducts [73] (Figure 3).

Aldosterone is known to increase ENaC activity and sodium reabsorption in the eccrine sweat glands [74]. Corin-mediated ANP production and function counter the aldosterone function. In WT mice, aldosterone treatment decreased sweat excretion, whereas such an effect was not observed in corin KO mice [73]. These results suggest that in WT mice, aldosterone-promoted salt reabsorption and corin-activated ANP-promoted salt excretion were in balance, which was tilted, by exogenous aldosterone, in favor of salt reabsorption. In contrast, corin KO mice lack the anti-aldosterone function. As a result, endogenous aldosterone has reached the maximal effect, which could not be further enhanced by exogenous aldosterone [73]. These results show that corin and ANP act as an anti-aldosterone mechanism in the skin to promote sodium and sweat excretion (Figure 3).

#### 2.2.2. Dermal Papilla and Coat Color in Animals

The dermal papilla is another site of corin expression. In mice of the *Agouti* background, corin deficiency renders a lighter coat color [45]. Genetic analyses indicate that corin is a suppressor of the agouti pathway in coat color specification, and that this function requires the protease activity of corin [45,75]. Consistently, *Corin* has been identified as one of the three major pigmentation genes in beach mice in the Gulf and Atlantic Coasts of the United States [76].

Similarly, a *CORIN* variant, causing H587Y substitution in the LDLR6 repeat of corin, was found to be a modifier of the dark coat stripes in tigers [77]. Biochemical studies indicate that corin suppresses the activity of Agouti signaling protein (ASIP), which inhibits melanocortin binding to melanocorin-1 receptor (MC1R) in the production of dark pigments [77]. Decreased corin activity increases ASIP function, thereby reducing the darkness of coat stripes in tigers [77]. These results are supported by a recent report, in which a *CORIN* variant, causing R795C substitution in the scavenger receptor domain, was associated with the golden (lighter) coat phenotype in Siberian tabby cats [78]. It will be interesting to determine whether corin plays a similar role in coat color specification in other mammalian species.

In humans, corin is expressed in dermal stem and progenitor cells [79,80,81], and in hair follicles [73]. The significance of corin expression in human hair follicles remains unclear. There are no reports of *CORIN* variants associated with skin or hair color in humans. In chickens and sheep, *CORIN* is a genetic factor contributing to evolutionary adaptation in hot arid environments [82,83], probably reflecting the role of corin in promoting salt and water excretion [73]. Reduced corin activity is expected to increase salt and water retention, offering a survival advantage in hot arid environments. Consistently, a *CORIN* variant with reduced activity is found in individuals whose ancestry can be traced back to southern regions of the Sahara Desert [33,84]. In modern times, however, ample supply of water and dietary salt puts the individuals with the *CORIN* variant at a higher risk of developing hypertension and heart disease [33,85].

## 3. Mechanisms in Uterine Spiral Artery Remodeling

In pregnancy, the uterus undergoes major phenotypical changes, including endometrial decidualization and spiral artery remodeling, which are essential for embryo implantation and fetal growth [86]. As a result, significant transforming events occur in uterine cells, including endometrial stromal cells, vascular smooth muscle cells (SMCs), and endothelial cells (ECs). Eventually, many SMCs in uterine spiral arteries are lost, and ECs are substituted by invading placental trophoblasts [86]. The resultant spiral arteries are larger in diameter and less responsive to maternal hormonal regulation, allowing steadily increased uteroplacental blood flow with reduced velocity to support the growing fetus [87]. Impaired uterine decidualization and spiral artery remodeling are key contributing factors in preeclampsia, a major disease in pregnancy [88,89]. To date, the molecular mechanisms underlying the cellular events in uterine decidualization and spiral artery remodeling are not well defined.

Corin and ANP play an important role in vascular remodeling. In mice and humans, corin expression is upregulated in the pregnant uterus, where corin-activated ANP promotes spiral artery remodeling and trophoblast migration [9,90]. Recent comparative transcriptomic studies in mammalian species show that *CORIN* is expressed in a subset of endometrial stromal lineage cells, and may contribute to the evolution of deep placental invasion and extensive spiral artery remodeling [91]. Consistently, pregnant corin and ANP KO mice have delayed trophoblast invasion and poorly remodeled spiral arteries [9]. The mice also develop gestational hypertension and proteinuria, a preeclampsia-like phenotype [5,9].

To date, defective *CORIN* variants have been found in patients with preeclampsia [9,35,92]. Altered uterine and placental corin expression and shedding have also been reported in animal models and pregnant women with hypertension [9,93,94,95,96]. Moreover, increased plasma or serum corin levels are found in preeclamptic patients [9,97,98,99,100,101,102,103]. Further studies should help to determine the tissue origin (e.g., heart vs. uterus or placenta) of the detected circulating corin, and to understand whether the circulating corin levels reflect increased corin expression and/or shedding in the tissue under disease conditions.

The *CORIN* transcription in uterine cells differs from that in cardiomyocytes. Recently, Krüppel-like factor 17 (KLF17) has been identified as a key transcription factor in uterine *CORIN* expression [104]. In cultured human uterine endometrial cells, disruption of the *KLF17* gene prevented *CORIN* expression [104]. In *Klf17* KO mice, no *Corin* expression was detected in the uterus. Similar to corin KO mice, *Klf17* KO mice develop a preeclampsia-like phenotype in pregnancy [104]. In addition, other transcription factors, including progesterone receptor, GATA2, and nuclear receptor subfamily 2 group F member 2 (NR2F2), also play a crucial role in *CORIN* expression in human endometrial cells [91]. Consistently, progesterone treatment increased both *Corin* and *Klf17* expression in ovariectomized mice [104].

More recent studies have provided new insights into the molecular mechanisms underlying the uterine corin and ANP function. In experiments with mouse models and cultured human uterine cells, corin and ANP were found to promote sequential molecular and cellular events in uterine decidualization and spiral artery remodeling [105]. Particularly, uterine corin and ANP enhanced endometrial decidualization and the expression of TNF-related apoptosis-inducing ligand (TRAIL), a pro-apoptotic protein, in endometrial stromal cells. TRAIL secreted from the decidualized endometrial cells induced apoptosis in spiral artery SMCs, which in turn released cyclophilin B to upregulate TRAIL receptors in ECs, thereby causing TRAIL-mediated apoptosis in ECs [105]. The sequential loss of SMCs and ECs paves the way for placental trophoblast invasion into uterine spiral arteries (Figure 4).

In support of this idea, impaired uterine decidualization and TRAIL expression were found in corin and ANP KO mice [105]. Depletion of TRAIL from endometrial stromal cell-derived conditional medium prevented apoptosis in cultured human uterine SMCs [105]. Moreover, SMC-derived or recombinant cyclophilin B induced TRAIL receptor expression in human uterine ECs in culture, leading to TRAIL-induced apoptosis in ECs [105]. These findings delineate the uterine corin and ANP function in sequential molecular and cellular events in uterine decidualization and spiral artery remodeling, which are important for healthy pregnancies. Additional studies are needed to verify these findings, and to test whether defects in the corin and ANP-induced TRAIL pathway contribute to preeclampsia in humans.

## 4. Therapeutic Potential in Heart Failure

In addition to the systemic effect of lowering blood volume and pressure, ANP has a direct anti-hypertrophic and anti-inflammatory function in the heart [1,13]. Consistently, cardiac hypertrophy has been observed in ANP and corin KO mice [43,84,106]. In pregnant corin KO mice, gestational hypertension is associated with cardiac hypertrophy, a phenotype resembling peripartum cardiomyopathy in human patients [107]. In cultured cardiac myocytes, corin overexpression prevented oxidative stress-induced cell death via a mechanism mediated by PI3K/AKT and NF-κB signaling [108]. To date, many *CORIN* variants have been reported in patients with hypertension, atrial fibrillation, coronary artery disease, and heart failure (HF) [29,109,110]. Further studies should help us to understand the functional significance of those *CORIN* variants in specific cardiovascular diseases.

HF is a serious disease, leading to the progressive loss of cardiac output. At late stages of HF, patients suffer from shortness of breath, orthopnea, and ankle swelling, which reflects poor circulation, lung congestion, and body fluid retention. In HF patients, high levels of plasma pro-ANP and pro-BNP are common, indicating an underlying deficiency in natriuretic peptide activation [111,112], which likely contributes to impaired body fluid homeostasis. Consistently, reduced corin expression and/or activity are associated with impaired natriuretic peptide activation and compromised cardiac function in HF patients [85,113,114]. Moreover, low levels of plasma or serum corin are found in HF patients with worse clinical outcomes [115,116,117,118,119,120,121,122]. These results suggest that corin deficiency may be an underlying mechanism in the pathogenesis of HF.

To date, natriuretic peptides are used as therapeutic agents to reduce body fluid retention and improve cardiac function in HF patients [123,124,125]. Recombinant ANP is also used to treat patients with renal failure [126]. In plasma, neprilysin-mediated cleavage is an important mechanism in natriuretic peptide degradation. High levels of plasma neprilysin have been reported in HF patients [120,122]. Inhibition of neprilysin increases natriuretic peptide levels in vivo [127]. This mechanism has been exploited to develop a new class of combined angiotensin receptor–neprilysin inhibitors (ARNIs) for reducing mobility and mortality in HF [128,129].

Given the role of corin in activating cardiac natriuretic peptides, therapeutic strategies may be considered to increase corin activity, and hence natriuretic peptide activation, in failing hearts. Indeed, corin overexpression in the heart improved cardiac function and prolonged survival in mouse models of cardiomyopathy and myocardial infarction [130,131]. Unexpectedly, overexpression of a mutant form of corin that lacked the catalytic activity enhanced cardiac function in mice with cardiomyopathy, indicating that corin may act through an alternative mechanism independent of its proteolytic activity [132]. More studies are required to define the biochemical basis of the potential non-catalytic function of corin.

Corin is a transmembrane protease. The N-terminal transmembrane domain is dispensable regarding ANP activation [133]. Possibly, a soluble form of corin (sCorin), lacking the cytoplasmic and transmembrane domains, could be used as a biological agent to enhance natriuretic peptide activity and cardiac function in HF. This hypothesis was tested recently in mouse models [134]. Injection of sCorin, either intravenously or intraperitoneally, resulted in readily detectable levels of sCorin in plasma, with half-lives of >3 h and >8 h, respectively [134]. In comparison, plasma half-lives of ANP and BNP in healthy individuals and HF patients ranged from <10 min to >20 min [135,136,137]. The short plasma half-life is one of the limitations of natriuretic peptides in clinical use. In sCorin-treated mice, plasma ANP and BNP levels were significantly increased. The mice also had higher levels of cyclic guanosine monophosphate (cGMP) in plasma and heart tissues, and lower levels of plasma angiotensin II and aldosterone, compared to those in vehicle-treated mice [134]. These results indicate that sCorin was active in vivo, which enhanced natriuretic peptide activation and signaling and suppressed the renin–angiotensin–aldosterone system. Importantly, sCorin treatment improved cardiac morphology and function in mouse models of HF induced by left coronary artery ligation and transverse aortic constriction, respectively [134]. These results are promising, showing the feasibility of recombinant corin as a biological agent in treating HF.

## 5. Conclusions and Perspectives

It has been more than two decades since the cloning of corin from the human heart [22]. We now know that corin is a key protease in the natriuretic peptide system that preserves normal blood pressure and cardiac function. Latest results show that corin acts not only in the heart, but also in non-cardiac tissues, such as kidney and skin tissues, to regulate salt excretion and body fluid homeostasis. In the pregnant uterus, corin and ANP mediate dynamic interactions among endometrial stromal and vascular cells to ensure an orderly spiral artery remodeling process. Studies using mouse models also indicate that corin-based biological agents may be developed to treat HF. We anticipate the requirement for more investigations to uncover additional corin substrates and/or functions, and to understand how corin functions are regulated in tissue-specific settings. We also expect that additional genetic studies will be needed to elucidate the impact of *CORIN* variants on cardiovascular homeostasis under physiological and pathological conditions. Finally, we envision that the knowledge gained regarding the biology of corin will be translated into diagnostic and/or therapeutic agents in the future to benefit patients with cardiovascular disease.

## Figures and Tables

**Figure 1 biology-11-00717-f001:**
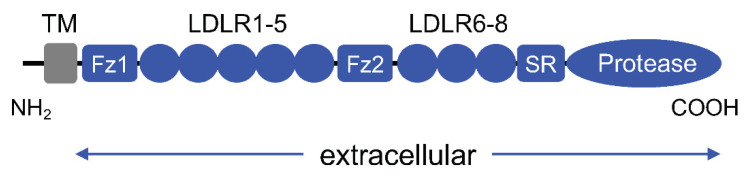
Schematic presentation of corin protein domains. Corin consists of an N-terminal (NH_2_) cytoplasmic tail, a single-span transmembrane domain (TM), and an extracellular region (indicated) that includes two frizzled domains (Fz1 and Fz2), eight low-density lipoprotein receptor-like repeats (LDLR1-8), a scavenger receptor-like domain (SR), and a trypsin-like serine protease domain at the C-terminus (COOH).

**Figure 2 biology-11-00717-f002:**
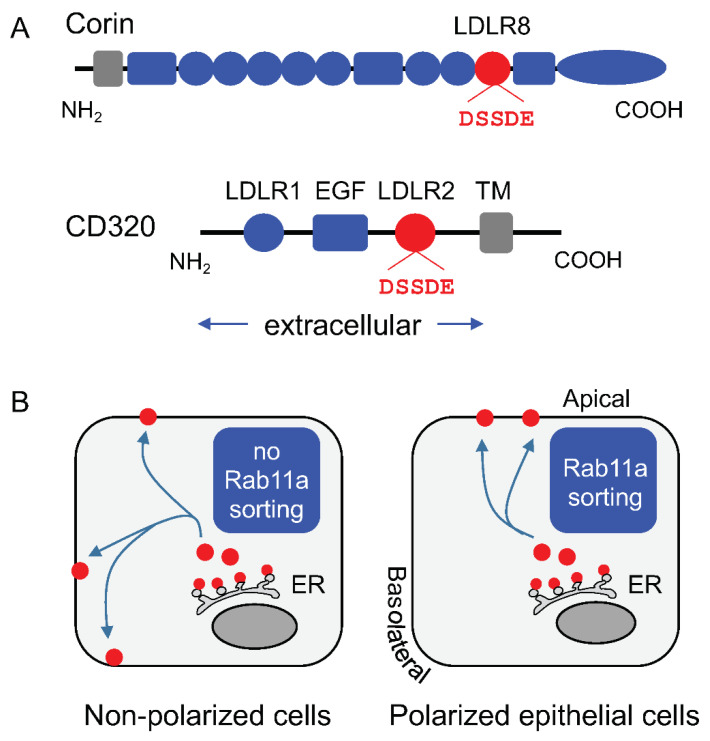
Apical membrane targeting of corin and CD320 in polarized epithelial cells. (**A**) Corin protein domains are described in the legend of Figure 1. CD320 consists of an N-terminal extracellular region with two LDLR repeats (LDLR1 and LDLR2) and an epidermal growth factor-like domain (EGF), a transmembrane domain (TM), and a C-terminal cytoplasmic tail. A DSSDE amino acid motif (red) is present in the LDLR8 domain in corin and LDLR2 domain in CD320. (**B**) In non-polarized cells without the Rab11a-mediated apical sorting mechanism, corin and CD320 (red dots) synthesized in the endoplasmic reticulum (ER) are distributed to the entire cell surface (left panel). In polarized epithelial cells with the Rab11a-mediated apical sorting mechanism, the DSSDE motif serves as an apical sorting signal for corin and CD320 expression on the apical, but not basolateral, membrane (right panel).

**Figure 3 biology-11-00717-f003:**
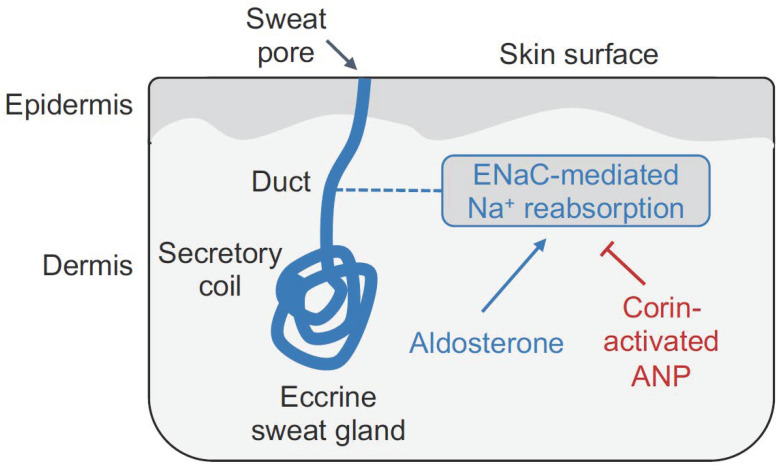
A proposed role of corin and ANP in the eccrine sweat gland. The eccrine sweat gland in the skin consists of the secretory coil, where initial sweat is produced, and the duct, where ENaC-mediated Na^+^ reabsorption occurs. Aldosterone promotes ENaC-mediated Na^+^ reabsorption. In contrast, corin and ANP inhibit ENaC-mediated Na^+^ reabsorption, thereby increasing Na^+^ and sweat excretion.

**Figure 4 biology-11-00717-f004:**
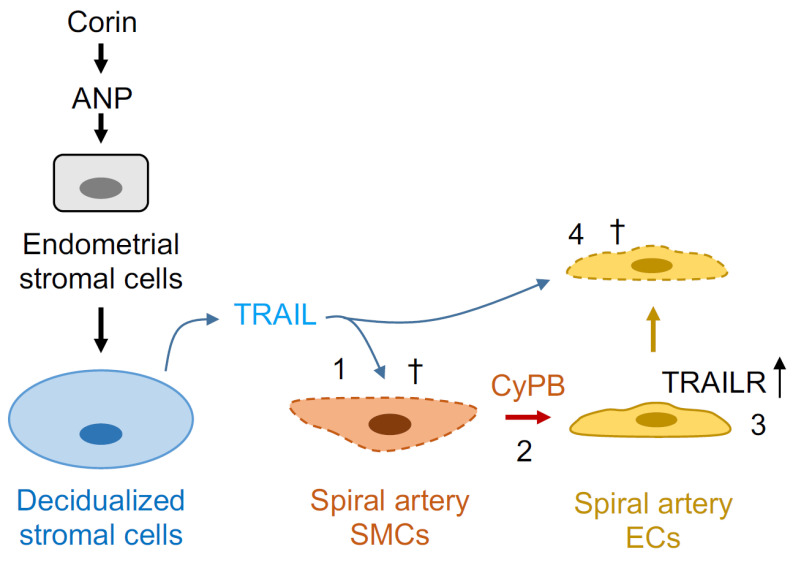
A proposed mechanism of corin and ANP in uterine spiral artery remodeling. In pregnancy, corin is expressed in the uterus to activate ANP, which promotes decidualization of uterine stromal cells and TNF-related apoptosis-inducing ligand (TRAIL) expression. TRAIL released from decidualized stromal cells induces apoptosis (†) in spiral artery smooth muscle cells (SMCs) (1) and cyclophilin B (CyPB) release (2). CyPB, in turn, upregulates TRAIL receptor (TRAILR) expression in spiral artery endothelial cells (ECs) (3), thereby causing TRAIL-mediated apoptosis in ECs (4).

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
