# Peer review of "Corin: A Key Mediator in Sodium Homeostasis, Vascular Remodeling, and Heart Failure"

_biology, 2022, doi:10.3390/biology11050717_

Round 1
Reviewer 1 Report
This is a review article by Zang et al., where possible roles of corin, a
transmembrane enzyme activating ANP, in non-cardiac tissues including
kidneys and uterus are discussed, and in addition, therapeutic potential
for this enzyme in heart failure is mentioned. This is an interesting
paper, focusing on the roles of corin-ANP in non-cardiac tissue, and
well-written by citing research papers recently published. A couple of
minor comments can be raised for authors’ consideration.
1. Figures 2B represent non-polarized (left) and polarized epithelial
cells (right). In the latter figure, the word “Apical” is put on the
top, while “Basolateral” on the left side. This reviewer wonders if
“Basolateral” should be put on the bottom as well.
2. Anatomy or physiology of eccrine glands of the skin might be unfamiliar
for readers specializing in cardiac bioactive peptides. It should be nice
for those readers if the authors can provide any figure in describing the
role of corin and ANP in eccrine glands of the skin.
Author Response
- Figures 2B represent non-polarized (left) and polarized epithelial cells (right). In the latter figure, the word “Apical” is put on the top, while “Basolateral” on the left side. This reviewer wonders if “Basolateral” should be put on the bottom as well.
Response: We thank the reviewer for the suggestion and have revised Figure 2B by placing “Basolateral” at the low left corner of the diagram.
- Anatomy or physiology of eccrine glands of the skin might be unfamiliar for readers specializing in cardiac bioactive peptides. It should be nice for those readers if the authors can provide any figure in describing the role of corin and ANP in eccrine glands of the skin.
Response: This is a valuable point. We have added a figure (new Figure 3) to illustrate the proposed role of corin and ANP in the eccrine sweat glands.

Reviewer 2 Report
To the Authors
General Considerations
The aim of this study was to discuss latest findings regarding the molecular and cellular mechanisms underlying the role of corin in sodium homeostasis, uterine spiral artery remodeling, and heart failure. This is an interesting review. I have also some specific points to address to the Authors with the aim to further improve the scientific message of this article.
Specific Points
- Simple Summary, line 11. The sentence: “the ANP is a vital hormone” may be misleading. The presence at least one of the two genes natriuretic peptides (ANP or BNP) is “vital” (i.e., essential for life). Even the complete loss of biological function of a single gene ANP (or BNP) is not fatal. This experimental evidence probably explains because there are 3 genes related to Natriuretic Peptides in mammals (ANP, BNP and CNP). The 3 natriuretic peptide-related genes have a common gene ancestor: the VNP gene (Inoue K et al. Mol Biol Evol 2005;22:2428-2434). Some gene variants of ANP and BNP genes are strictly related to some cardiovascular diseases, while others variants may have even a more favorable metabolic profile (Vinnakota S, Chem HH. J Endocrinol Soc 2020;4(6):bvaa052.
- Plasma half-life of ANP and BNP, line 302-304. Plasma half-life of ANP in healthy adult subjects is < 10 min, but that of BNP is > 20 min (Clerico A et al. Curr Drug Metab 2000;1:85-105). These different in vivo kinetics of the ANP and BNP should be considered in the pathophysiological and clinical interpretations of circulating levels of different natriuretic peptides in healthy subjects and patient with cardiovascular diseases (Clerico A et al. clin Chem Lab Med 2006;44:366-378).
- Therapeutic Potential in Heart Failure. Authors should discuss the possible variations in activity/plasma concentration of corin in patients with heart failure in treatment with drugs able to affect the production/degradation of natriuretic peptdes (in particular the new ARNI drugs) (Zaidi SS et al. Biomed Res Int 2018;2018:7279036; Meyre PL: et al. JAAC HF 2021;9:96-99; Aimo A et al. Crit Rev Lab Sci 2021;58:530-545).
Author Response
Reviewer 2
The aim of this study was to discuss latest findings regarding the molecular and cellular mechanisms underlying the role of corin in sodium homeostasis, uterine spiral artery remodeling, and heart failure. This is an interesting review. I have also some specific points to address to the Authors with the aim to further improve the scientific message of this article.
Specific Points
- Simple Summary, line 11. The sentence: “the ANP is a vitalhormone” may be misleading. The presence at least one of the two genes natriuretic peptides (ANP or BNP) is “vital” (i.e., essential for life). Even the complete loss of biological function of a single gene ANP (or BNP) is not fatal. This experimental evidence probably explains because there are 3 genes related to Natriuretic Peptides in mammals (ANP, BNP and CNP). The 3 natriuretic peptide-related genes have a common gene ancestor: the VNP gene (Inoue K et al. Mol Biol Evol 2005;22:2428-2434). Some gene variants of ANP and BNP genes are strictly related to some cardiovascular diseases, while others variants may have even a more favorable metabolic profile (Vinnakota S, Chem HH. J Endocrinol Soc 2020;4(6):bvaa052.
Response: We thank the reviewer for this valid point. Accordingly, we have revised the sentence by replacing the word “vital” with “important” (Simple Summary, line 1). To avoid redundant words, we have also revised the wording in the Simple Summary and Abstract, as highlighted in red. In addition, we have added the reference of Inoue K et al. (new ref. #60).
- Plasma half-life of ANP and BNP, line 302-304. Plasma half-life of ANP in healthy adult subjects is < 10 min, but that of BNP is > 20 min (Clerico A et al. Curr Drug Metab 2000;1:85-105). These different in vivo kinetics of the ANP and BNP should be considered in the pathophysiological and clinical interpretations of circulating levels of different natriuretic peptides in healthy subjects and patient with cardiovascular diseases (Clerico A et al. Clin Chem Lab Med 2006;44:366-378).
Responses: Indeed, reported human ANP and BNP plasma half-lives vary, depending on methods used and health conditions of the studied individuals. We have revised the sentence to “In comparison, plasma half-lives of ANP and BNP in healthy individuals and HF patients ranged from <10 min to >20 min (135-137)”. Two references by Clerico A, et al. have been added (new refs. 136 and 137) (page 15, second paragraph).
- Therapeutic Potential in Heart Failure. Authors should discuss the possible variations in activity/plasma concentration of corin in patients with heart failure in treatment with drugs able to affect the production/degradation of natriuretic peptdes (in particular the new ARNI drugs) (Zaidi SS et al. Biomed Res Int 2018;2018:7279036; Meyre PL: et al. JAAC HF 2021;9:96-99; Aimo A et al. Crit Rev Lab Sci 2021;58:530-545).
Response: We thank the reviewer for the suggestion. We have revised the text to include the following: “In plasma, neprilysin-mediated cleavage is an important mechanism in natriuretic peptide degradation. High levels of plasma neprilysin have been reported in HF patients (120, 122). Inhibition of neprilysin increases natriuretic peptide levels in vivo (127). This mechanism has been exploited to develop a new class of combined angiotensin receptor-neprilysin inhibitors (ARNIs) for reducing mobility and mortality in HF (128, 129)” (page 14, second paragraph). The papers by Zaidi SS, et al. (ref. #122) and Mehre PL, et al. (ref. #120), Aimo A, et al. (ref. #129) and two additional papers (refs. 127 and 128) are cited.

Round 2
Reviewer 1 Report
They responded to my question appropriately, therefore, I have no other questions.
Reviewer 2 Report
Authors revised the manuscript in accordance with the suggestions made by the Reviewer. The scientific message of the study is now significantly improved.